# Predicting the Efficacy and Safety of TACTICs (Tumor Angiogenesis-Specific CAR-T Cells Impacting Cancers) Therapy for Soft Tissue Sarcoma Patients

**DOI:** 10.3390/cancers12102735

**Published:** 2020-09-23

**Authors:** Kento Fujiwara, Shigemi Sasawatari, Sho Nakai, Keisuke Imaeda, Seina Nagai, Yoshihiro Matsuno, Kanako Hatanaka, Yutaka Hatanaka, Satoshi Takenaka, Naoki Okada

**Affiliations:** 1Project for Vaccine and Immune Regulation, Graduate School of Pharmaceutical Sciences, Osaka University, Suita 565-0871, Japan; papaian0720@gmail.com (K.F.); imaeda-k@phs.osaka-u.ac.jp (K.I.); nagai-s@phs.osaka-u.ac.jp (S.N.); 2MEDINET Co. Ltd., Tokyo 143-0006, Japan; sasawatari-s@medinet-inc.co.jp; 3Department of Orthopaedic Surgery, Graduate School of Medicine, Osaka University, Osaka 565-0871, Japan; s.nakai.0925@gmail.com (S.N.); s.takenaka.0816@gmail.com (S.T.); 4Department of Orthopaedics, Nozaki Tokusyukai Hospital, Daito 574-0074, Japan; 5Department of Surgical Pathology, Hokkaido University Hospital, Sapporo 060-8648, Japan; ymatsuno@med.hokudai.ac.jp; 6Research Division of Companion Diagnostics, Hokkaido University Hospital, Sapporo 060-8648, Japan; kyanack@huhp.hokudai.ac.jp (K.H.); yhatanaka@huhp.hokudai.ac.jp (Y.H.); 7Clinical biobank, Clinical Research and Medical Innovation Center, Hokkaido University Hospital, Sapporo 060-8648, Japan; 8Department of Orthopaedics, Osaka International Cancer Institute, Osaka 541-8567, Japan

**Keywords:** TACTICs (tumor angiogenesis-specific CAR-T cells impacting cancers) therapy, anti-VEGFR2 CAR, mRNA electroporation, soft tissue sarcoma

## Abstract

**Simple Summary:**

Sarcomas have few effective treatment options due to the rarity and diversity and have a high risk of recurrence and metastasis. Therefore, the development of new therapeutics that can meet their medical needs is required. Our adoptive immunotherapy strategy using T cells to express the chimeric antigen receptor (CAR) against vascular endothelial growth factor receptor 2 (VEGFR2), which is highly expressed on tumor vascular endothelial cells, has the potential to be a novel treatment against diverse sarcomas with abundant vascular invasion. Here, we optimized the manufacturing and transportation of anti-VEGFR2 CAR-mRNA-transfected T cells and collected information that allowed the extrapolation of their efficacy and safety potential for sarcoma patients. Our results support the development of a “first in humans” study to evaluate the potential of our anti-VEGFR2 CAR-T cell therapy as a new treatment option for sarcoma patients.

**Abstract:**

Soft tissue sarcomas (STSs) are heterogeneous and aggressive malignancies with few effective therapies available. We have developed T cells expressing a vascular endothelial growth factor receptor 2 (VEGFR2)-specific chimeric antigen receptor (CAR) to establish a tumor angiogenesis-specific CAR-T cells impacting cancers (TACTICs) therapy. In this study, we optimized the manufacturing and transportation of mRNA-transfected anti-VEGFR2 CAR-T cells and collected information that allowed the extrapolation of the efficacy and safety potential of TACTICs therapy for STS patients. Although 5-methoxyuridines versus uridines did not improve CAR-mRNA stability in T cells, the utilization of CleanCap as a 5′ cap-structure extended the CAR expression level, increasing VEGFR2-specific cytotoxicity. Furthermore, 4 °C preservation conditions did not affect the viability/cytotoxicity of CAR-T cells, contrarily to a freeze-thaw approach. Importantly, immunohistochemistry showed that most of the STS patients’ specimens expressed VEGFR2, suggesting a great potential of our TACTICs approach. However, VEGFR2 expression was also detected in normal tissues, stressing the importance of the application of a strict monitoring schedule to detect (and respond to) the occurrence of adverse effects in clinics. Overall, our results support the development of a “first in humans” study to evaluate the potential of our TACTICs therapy as a new treatment option for STSs.

## 1. Introduction

Sarcomas arising from soft tissues throughout the body, such as muscle, fat, and nerves, exhibit a variety of histological types, depending on their tissue of origin and degree of differentiation; their incidence is very low compared to solid tumors of epithelial tissue origin [1,2,3]. Surgery is the primary option for the treatment of soft tissue sarcomas (STSs), with high survival rates expected after extensive resection. However, highly invasive surgical resection leads to poor postoperative quality of life and has a significant impact on patients’ physical, mental, and career development. Of note, children and young adults have a high incidence of STSs. Furthermore, due to the high hematogenous metastatic potential of sarcoma cells, patients with STSs are forced to live with the fear of recurrence and the appearance of distant metastasis, even if the primary tumor is resected [4]. Importantly, because of the lack of effective drugs for patients manifesting recurrence/metastasis and for patients with non-surgically resectable STSs, the development of new therapeutics that can meet their medical needs is desperately required.

In recent years, novel cancer treatments such as immune checkpoint inhibitors and T-cell adoptive immunotherapy have demonstrated drastic therapeutic benefits for epithelial solid tumors [5,6,7]. Although it is desirable to expand the application of these immunotherapies to STSs, their effectiveness is only limited to some, such as undifferentiated pleomorphic sarcoma, alveolar soft part sarcoma, and synovial sarcoma [8,9,10,11]. This is not only due to the fact that most STSs are cold tumors lacking tumor-associated antigens, which are the main targets of anti-cancer immune cells, but also because generally the tumor tissues are not infiltrated with anti-cancer immune cells [12]. On the other hand, as in epithelial carcinomas, STSs induce angiogenesis from nearby parent vessels as the tumor grows, and vascular endothelial cells comprising these tumor vessels have been shown to strongly express vascular endothelial growth factor receptor 2 (VEGFR2) [13,14]. Previously, we have developed adoptive immunotherapy using T cells expressing a chimeric antigen receptor (CAR) that targets VEGFR2 and reported that anti-VEGFR2 CAR-T cells could inhibit tumor growth based on tumor vascular injury in a variety of tumor-bearing mouse models [15,16]. Our TACTICs (Tumor Angiogenesis-specific CAR-T cells Impacting Cancers) therapeutic strategy has the potential to be an all-around effective treatment against diverse STSs and to prevent hematogenous metastases via vascular injury. In fact, since it has been reported that cancer cells from some cases of angiosarcoma and synovial sarcoma express VEGFR2 [14,17], TACTICs therapy might demonstrate excellent tumor elimination by direct cytotoxicity towards the sarcoma cells. Therefore, we considered conducting a first in humans (FIH) study to evaluate the safety and efficacy of TACTICs therapy as a new treatment option for refractory STSs.

In every FIH study, ensuring the safety of the subjects is of the highest priority. CAR-T cells can be produced by gene transfer using viral vectors and provide long-term antitumor effects with a single dose; however, their use carries the risk of genotoxicity. Therefore, the safety management of subjects should be carried out continuously and consistently, as the complete elimination of transferred CAR-T cells from the body can be difficult when signs of serious adverse effects are observed [18]. Therefore, we selected electroporation (EP) transfer of mRNA encoding the CAR gene as a method for generating CAR-T cells for clinical research [16]. The mRNA-EP method can reduce the risk of side effects by allowing the reversion of CAR-T cells to their original phenotype over time; however, in this context, the efficacy of CAR-T cells is obviously transient. A previous study has confirmed that the mRNA-EP method was an efficient way to express CAR on T cells without affecting their viability and functions. Further, the study revealed that multiple doses of anti-VEGFR2 CAR-mRNA-transfected T cells exhibited strong anti-tumor activity, comparable to that provided by a single dose of CAR-T cells generated via viral vector-mediated gene transfer [16].

In the preparation for the FIH study using mRNA-transfected CAR-T cells, we needed to determine the final development candidate (the specific anti-VEGFR2 CAR-T cells batch) and establish a method for producing and transporting 10^6^–10^10^ CAR-T cells to be administered in a single dose. In particular, the mRNA construct optimization for T cells’ transfection, to ensure a favorable CAR expression profile on their membranes, may improve the intensity and duration of CAR-T cells’ function [19,20]. In addition, removal of non-self-sequences in the CAR extracellular region as much as possible is expected to reduce the antigenicity of CAR-T cells and diminish their potential elimination by the host immune cells [21]. Importantly, to properly evaluate the efficacy and safety of TACTICs therapy in a small population, it is necessary to select patients who are expected to respond well to this therapy. While, as abovementioned, angiosarcoma, synovial sarcoma, liposarcoma, and undifferentiated polymorphic sarcoma have been reported to strongly express VEGFR2, and are expected to be excellent targets for TACTICs therapy, not all STS cases are VEGFR2 positive [13,14]. Therefore, the development of companion diagnostics and the establishment of biomarkers are desirable to select VEGFR2-positive cases, potentially good responders to our anti-VEGFR2 CAR-T cell-based therapy.

In this study, we produced anti-VEGFR2 CAR-T cells for clinical research using a GMP-compatible EP-enabled electroporator. We optimized the CAR-mRNA construct to obtain the final development candidate for the anti-VEGFR2 CAR-T cell-based therapy. In addition, we selected the optimal transport conditions for mRNA-transfected CAR-T cells, from the cell manufacturing facility to the centers where STS patients are expected to be treated, in a way to maintain CAR-T cell functions. In addition, we collected information that allowed the selection of STS cases suitable for the future application of our TACTICs therapy: we predicted the potential efficacy and safety of our approach using immunohistochemical analysis of human STSs-derived and normal tissues.

## 2. Results

### 2.1. Optimization of CAR-mRNA for the Development of the Final CAR-T Cell-Based Therapy

In a preliminary study using two second-generation CARs incorporating anti-VEGFR2 single-chain variable fragments (scFv), CAR [V/28/28/28/28-3z] (CAR#1)-T cells showed a superior VEGFR2-specific cytotoxic activity compared to CAR [V/8a/8a/137-3z]-T cells; of note, the latter showed higher intensity and longer duration in CAR expression (Appendix A). Thus, we selected CAR#1-T cells as an anti-VEGFR2 CAR-T cell-based therapy prototype, expected to be highly effective in the context of clinical use. Subsequently, aiming to reduce the immunogenicity of CAR-T cells and to improve the persistence of their anti-tumor activity, we produced CAR#2-coding mRNA from which the tag sequences for CAR protein analysis were removed from the conventional CAR#1-coding mRNA; CAR#3-coding mRNA, which was CAR#2-coding mRNA codon-optimized for the human codon usage preference; CAR#4-coding mRNA, derived from CAR#3-coding mRNA whose 5′ cap structure was changed from the anti-reverse cap analog (ARCA) to the CleanCap; and CAR#5-coding mRNA, derived from CAR#4-coding mRNA, whose uridines were replaced by 5-methoxyuridines (Figure 1A). First, we analyzed the CAR expression profile in human T cells transfected with these different CAR-mRNAs using the EP method. All five CAR-mRNAs were efficiently introduced into T cells using EP. Furthermore, all of them showed a comparable decrease after EP and were diluted and degraded to detection limit levels after 48 h (Figure 1B). Removal of the tag sequence (CAR#2-coding mRNA) and codon optimization (CAR#3-coding mRNA) did not affect the cell surface expression profile of CAR proteins (Figure 1C). On the other hand, the modification of the 5′ cap structure from ARCA to CleanCap (CAR#4-coding mRNA) resulted in increased CAR expression levels and duration. Furthermore, the substitution of uridines by 5-methoxyuridines (CAR#5-coding mRNA) resulted in high CAR expression levels in the early post-EP period; however, the expression declined rapidly thereafter, with CAR expression levels at 48 h being as low as those of CAR#1-, #2-, and #3-coding mRNAs. These results revealed that the translation efficiency of CAR proteins could be increased by using CleanCap as the 5′ cap structure, whereas the stability of CAR-mRNA could not be improved by using modified uridines.

Next, the cytotoxicity of each CAR-mRNA-transfected T cell was assessed 24 h after EP (Figure 1D). All CAR-T cells specifically injured VEGFR2-expressing L1.2 cells (VEGFR2^+^ L1.2 cells). Remarkably, CAR#4- and CAR#5-T cells showed markedly higher cytotoxic activity than the other three CAR-T cells. This result directly reflects the CAR expression levels, suggesting that the modification of CAR-mRNA constructs does not affect the viability and characteristics of CAR-T cells. Based on the results of CAR expression profiles and transfected T cells’ cytotoxic activities, we determined that CAR#4-coding mRNA was the optimal CAR-mRNA construct for the generation of a final development candidate for anti-VEGFR2 CAR-T cell-based therapies, ensuring potent and sustained cytotoxic activity.

### 2.2. Investigation of the Optimal Transport Conditions for mRNA-Transfected CAR-T Cells

To administer CAR-T cells with high anti-tumor activity to patients, it is necessary to establish an efficient transporting method that prevents CAR-T cell damage and simultaneously maintains CAR expression levels as much as possible. First, we examined whether mRNA-transfected CAR-T cells could be cryopreserved and then transported. A portion of the CAR-T cells 3 h after EP transfection (fully recovered from the EP-derived cell membrane damage) was slowly frozen within a frost protection reagent (CP-1) and rapidly thawed 21 h later (24 h after EP transfection), while the remaining cells were kept in culture at 37 °C. Interestingly, the CAR expression levels in frozen CAR-T cells, immediately after thawing, were higher than those in the CAR-T cells cultured at 37 °C for 24 h; thereafter, CAR expression gradually decreased with time in both conditions, with CARs completely disappearing from the T cell membranes 72 h after thawing (96 h after EP transfection), or 72 h after transfection for cultured CAR-T cells (Figure 2A). Of note, CAR expression profiles were similar in both conditions, indicating that the freeze-thaw treatment did not affect the stability of intracellular CAR-mRNA. On the other hand, the cytotoxic activity of CAR-T cells against VEGFR2^+^ L1.2 cells, immediately after freeze-thawing, was significantly reduced, compared to that of cultured cells (Figure 2A, *p* < 0.01). Importantly, this phenotype was not a consequence of the loss of cell viability: >98%, similar to that detected for cultured CAR-T cells (data not shown). These results revealed that freeze-thaw manipulation highly impacted the effector function of T cells, even in the context of preserved CAR expression by T cells. Therefore, we concluded that cryopreservation and transport of mRNA-transfected CAR-T cells are not appropriate considering clinical applications, since CP-1 is the only freeze protection reagent approved for use in clinics.

Next, as an alternative to cryopreservation, we examined the effect of transport under 18 °C or 4 °C conditions on mRNA-transfected CAR-T cells. CAR-T cells, cultured at 37 °C for 18 h after EP, were subjected to the different temperature conditions for approximately 6 h, and their CAR expression levels and cytotoxic activity were assessed immediately after (24 h after EP; Figure 2B). Remarkably, CAR-T cells subjected to both temperatures showed equivalent high CAR expression levels and cytotoxic activity, comparable to those of CAR-T cells cultured at 37 °C for 24 h (Figure 1D and Figure 2B). Thereafter, the 4 °C environment, which is generally considered to be superior in maintaining T cell viability and effector function, was selected as the transport condition for tumor vessel-injuring CAR-T cells. Then, to validate the in vivo efficacy of anti-VEGFR2 CAR-T cells transported at 4 °C, we analyzed their elimination effect on VEGFR2^+^ L1.2 cells transferred to NOG mice (NOD/Shi-scid, IL-2RγKO Jic; Figure 2C). While the administration of mock-T cells did not induce any specific VEGFR2^+^ L1.2 cells’ elimination (as expected), CAR-T cells were able to eliminate 20% of the VEGFR2^+^ L1.2 cells within 2 days. These results suggested that transport of cells under 4 °C conditions would not impact CAR expression levels and cytotoxic activity of mRNA-transfected CAR-T cells.

### 2.3. Generation of an Antibody Specifically Recognizing the Relevant VEGFR2 Extracellular Epitope and Evaluation of Its Binding Properties

To properly select the STS cases eligible to receive the proposed TACTICs therapy, the development of companion diagnostic tools that recognize the same VEGFR2 extracellular region as our CAR-T cells is essential. Hence, we obtained a complete antibody possessing the variable region of anti-VEGFR2 scFv (clone V-85), the one used as the antigen recognition domain of CAR, and evaluated its binding properties to VEGFR2. The newly developed V-85 antibody was able to bind specifically to VEGFR2-expressing NIH/3T3 cells (VEGFR2^+^ NIH/3T3 cells) to a similar extent as that of a commercial anti-VEGFR2 antibody (clone 7D4-6) recognizing the VEGFR2 extracellular domain, as per both immunocytochemistry and flow cytometry analysis (Figure 3A). Moreover, to confirm whether the recognition epitope of the V-85 antibody in the VEGFR2 extracellular domain was indeed identical to that of anti-VEGFR2 CAR T cells, we examined the inhibitory effect of the V-85 antibody on the cell-cell binding of VEGFR2^+^ NIH/3T3 cells and anti-VEGFR2 CAR-expressing Jurkat cells (CAR^+^ Jurkat cells) (Figure 3B). Importantly, while the addition of 7D4-6 antibody showed a slight binding inhibition, the addition of V-85 antibody resulted in significantly greater inhibition, indicating that the V-85 antibody recognized the same epitope as our anti-VEGFR2 CAR.

### 2.4. Analysis of VEGFR2 Expression in Frozen Specimens of Human STS-Derived and Surrounding Normal Tissues

We subsequently performed a correlation analysis of VEGFR2 detection between the V-85 antibody and a commercial antibody recognizing the VEGFR2 intracellular region (clone 55B11) using immunohistochemistry in frozen specimens of STS-derived and normal tissues. Among 18 STS tissues obtained via wide local excision, VEGFR2 expression was positive in 14 (Figure 4A,B); of note, the VEGFR2 staining in 11 cases was consistent with the localization of the vascular endothelial cell marker CD31, regardless of which antibody was used (Figure 4A). In malignant peripheral nerve sheath tumor (MPNST) tissue from patient OU008, the staining with the 55B11 antibody was restricted to endothelial cells, whereas the staining with the V-85 antibody was found not only in endothelial cells but also in perivascular cells (Figure 4A,B). In angiosarcoma tissue from patient OU012, staining with the 55B11 antibody was limited to CD31-positive vascular endothelial cells, whereas the V-85 antibody stained only tumor cells, and not endothelial cells (Figure 4A,B). In addition, in myxoid liposarcoma tissue from patient OU006, slit-shaped VEGFR2 staining was observed in CD31-negative regions with both antibodies (Figure 4A,B). Although the VEGFR2 staining pattern between V-85 and 55B11 antibodies differed in some STS tissues, the VEGFR2 staining levels (ΔMFI per field) given using the V-85 antibody positively correlated with those given using the 55B11 antibody, and additionally, with CD31 staining levels (Figure 4C). These results show that STSs, with many infiltrating tumor vessels, are promising therapeutic targets for our anti-VEGFR2 CAR-T cells. Furthermore, the V-85 antibody may be useful as a companion diagnostic agent for selecting cases that would benefit from our envisioned TACTICs therapy.

Importantly, we also analyzed the expression of VEGFR2 in normal muscle, adipose, and skin tissues (surrounding STS tissues) to assess the safety of anti-VEGFR2 CAR-T cells therapy. Under the same observation conditions as STS tissues, muscle and adipose tissues did not show specific immunofluorescence staining with either anti-VEGFR2 antibody. On the other hand, the VEGFR2-stained images in skin tissues showed positive staining in the eccrine sweat glands when the 55B11 antibody was used; however, the same was not observed when we used the V-85 antibody. These results suggested that our anti-VEGFR2 CAR-T cells would not harm, at least, normal muscle, adipose, and skin tissues and could specifically injure tumor vascular endothelial cells in adjacent STS tissues.

### 2.5. Analysis of VEGFR2 Expression in Paraffin-Embedded Specimens from Rare Cancers and Major Organs

To investigate the possibility of expanding the indication of our TACTICs approach for solid tumors other than STSs, and to investigate the possible occurrence of adverse events in major organs other than muscle, fat, and skin, we analyzed VEGFR2 expression in paraffin-embedded rare cancer and normal tissue specimens. We used the 55B11 antibody that showed a positive correlation with the V-85 antibody with respect to VEGFR2 staining, since the V-85 antibody was unfortunately unable to detect VEGFR2 in paraffin-embedded specimens, after protein denaturation. Remarkably, VEGFR2 staining with the 55B11 antibody was observed in more than half of the rare cancer specimens (Figure 5A), suggesting that TACTICs therapy could be effective against many types of solid tumors. In particular, prostate cancer, gastrointestinal stromal tumor, malignant mesothelioma, and papillary thyroid cancer were found to be potential targets for TACTICs therapy, as all representative specimens showed VEGFR2-positive tumor vessels. Furthermore, even in normal tissues, staining with the 55B11 antibody was found in the kidneys, particularly in glomeruli, and in the parenchyma and glandular tissues from the prostate, colon, stomach, and the urothelial system (Figure 5B), indicating that caution should be exercised when administering anti-VEGFR2 CAR-T cells to humans, as these organs/systems might be associated with some adverse reactions.

## 3. Discussion

In this study, we first examined the specifications and transport conditions for the final development of a promising anti-VEGFR2 CAR-T cells candidate, in preparation for a future FIH study using TACTICs therapy. CAR-T cells generated via viral vectors are expected to have long-term anti-tumor effects due to persistent CAR expression; of note, CD137-derived STD are often incorporated into the second-generation CAR constructs to improve the in vivo persistence of CAR-T cells [22,23]. On the other hand, in CAR-T cells generated via mRNA-EP, memory differentiation and long-term survival of the transfected T cell are not as important, since they lose their CAR-mediated antigen specificity with time. In this study, we followed the second approach. In order to clearly support the mRNA-EP-derived CAR-T cells’ therapeutic potential (considering frequent administrations), it is essential to show that mRNA-transfected CAR-T cells exert a strong cytotoxic activity, at least in a short period of time. Indeed, here we determined that a second-generation CAR construct with a CD28-derived STD was suitable for producing our anti-VEGFR2 CAR-T cells. Of note, the nonspecific cytotoxic activity of CAR [V/8a/8a/137-3z]-T cells as shown in Appendix A suggested that the second-generation CAR construct incorporating CD137-derived STD might have a high risk of causing off-target toxicity. Therefore, the design of CAR constructs targeting tumor-associated antigens which are also slightly expressed in normal tissues requires further validation regardless of the method of CAR-T cells production.

The anti-tumor activity of mRNA-transfected CAR-T cells depends on the CAR expression levels and duration. Importantly, CAR expression profiles are supposedly determined by the persistence of CAR-mRNA and its translation efficiency (rate). The five CAR-mRNAs examined in this study showed a similar mRNA elimination profile in the T cells, suggesting that the difference in the cell surface expression profiles of CAR#1–#5 was attributed to the amount and rate of synthesis of CAR proteins (translation efficiency). The ribosomes initiate translation via recognition of the cap structure at the 5′ end of mRNA. CleanCap is known to have a higher capping efficiency to in vitro transcribed RNA than ARCA [19]. Therefore, we believed that the improvement in CAR expression levels due to the alteration of the 5′ cap structure from ARCA to CleanCap was caused by the increasing amount of translatable CAR-mRNA to which the 5′ cap structure was properly added. In addition, mRNAs synthesized using 5-methoxyuridine were reported to be less immunogenic and more efficient in translation [20]. Although our mRNA coding for CAR#5 did not improve intracellular persistence compared with the coding for CAR#4, CAR#5 showed the highest expression levels early after transfection. These results suggested that the translation rate of CAR#5-coding mRNA is superior to that of CAR#4-coding mRNA containing wild-type uridines. On the other hand, fast translation rates are known to reduce the folding efficiency of nascent proteins [24]. To be expressed as a functional protein, the nascent protein must be properly folded during biosynthesis in the endoplasmic reticulum and Golgi apparatus; therefore, the protein’s folding efficiency is also regulated by the rate of mRNA translation [24]. Although the details explaining why the initial expression of CAR#5 could not be maintained are not clearly known, we speculate that the rapid synthesis of artificial CAR#5 proteins may have led to the accumulation of unfolded proteins within the cells, resulting in their elimination (as aberrant proteins).

In addition, a series of experiments on CAR-mRNA construct modification revealed that the removal of the tag sequences in the CAR extracellular domain did not affect the efficiency of CAR expression and activation. Reducing the antigenicity of the CAR extracellular domain to the host immune system is an important consideration, as the appearance of human anti-mouse antibody (HAMA) and human anti-CAR antibody (HACA) can diminish the efficacy of CAR-T cell therapy [25]. On the contrary, our anti-VEGFR2 scFv (V-85), which was incorporated as the CAR extracellular domain, is a mouse-derived sequence that has not been humanized. Although we believe that scFv should be humanized to completely eliminate the antigenicity of CAR-T cells, the humanization of scFv carries the risk of altering the efficiency of CAR surface expression and antigen recognition properties [26]. Additionally, clinical trials using non-humanized anti-mesothelin CAR-mRNA transfected CAR-T cells suggested that the appearance of HACA, but not HAMA, limits the efficacy of CAR-T cell therapy. Hence, in the present study, we sought to improve the CAR-mRNA constructs to limit the risk of HACA appearance [21]. Needless to say, we plan to measure HACA and HAMA against CAR#4-T cells as biomarkers to monitor the efficacy and safety of the TACTICs therapy in the context of the FIH study.

The efficiency of T cell expansion or the functional strength of CAR-T cells is expected to differ depending on the background of individual patients; parameters such as age, complications, and treatment history may influence TACTICs therapy efficacy. Therefore, we have to confirm beforehand if the prepared mRNA-transfected CAR-T cells meet the expected function. Cryopreservation not only allows stockpiling and transporting of functional CAR-T cells, but also offers significant advantages for the assessment of the quality and function of prepared CAR-T cells over time, and immediately before they are administered to patients. In addition, multiple frozen stocks of the same lot allow for the continuous administration of CAR-T cells with consistent quality. However, since the effector function of CAR-T cells was greatly reduced by freeze-thaw procedures, cryopreservation would not be suitable to maintain our mRNA-transfected CAR-T cells. This, together with the notion that cryopreservation reagents available for clinical use are limited, led us to conclude that low-temperature transport (at 4 °C) is the best option for delivering mRNA-transfected CAR-T cells with retained function to patients. On the other hand, we cannot secure a grace period for CAR-T cells quality assessment under 4 °C transportation. Hence, it is necessary to determine the minimum-required parameters, such as viability, CAR expression levels, and CAR integrity, to predict the functionality of CAR-T cells in a short period, and to establish suitability criteria for administration to patients. Furthermore, future analyses using patient-derived T cells would be required to ensure the efficacy of mRNA-transfected anti-VEGFR2 CAR-T cells, since our study was limited to experiments using T cells derived from healthy volunteers.

The present analysis of VEGFR2 expression in frozen STS tissues suggested that TACTICs therapy would be a promising new treatment option for more than 70% of STS patients. Importantly, VEGFR2-positive cases were not characterized by a specific site of origin or STS type, and the levels of vascular invasion and VEGFR2 expression varied widely even within the same STS types. These facts emphasized again the importance of establishing companion diagnostic tools and biomarkers capable of quickly and appropriately selecting patients with VEGFR2-positive STSs, the ones that would benefit the application of TACTICs therapy. We believe that our V-85 antibody would be of great help in predicting the efficacy of TACTICs therapy beforehand. We showed that the V-85 antibody allowed the recognition of VEGFR2; of note, the V-85 antibody prevented the binding anti-VEGFR2 CAR-expressing cells, suggesting both share the same epitope. However, the V-85 antibody is not without limitations: it can only be used in frozen tissue sections, and not in paraffin-embedded ones. Still, and importantly, the commercially available 55B11 antibody showed a VEGFR2 staining potential that correlated with that of the V-85 antibody. Therefore, 55B11 is a useful companion diagnostic reagent applicable to paraffin-embedded biopsy specimens. However, having in mind that the VEGFR2 staining patterns comparing 55B11 and V-85 antibodies differed in some frozen tissue specimens, we cannot exclude possible under/overestimation of VEGFR2 expression in paraffin-embedded tumor and normal specimens. The 55B11 antibody has been widely used in VEGFR2 studies; many results of VEGFR2 expression analysis using this antibody in tumor and normal tissues have been reported [13,14,27]. In The Human Protein Atlas, 55B11 antibody-stained sections are available for different normal tissues, including kidney, colon, testis, skin, esophagus, and bladder tissues; of note, the stained images in tissues other than renal glomeruli were judged to be non-specific responses [27]. Although the epitope targeted by the 55B11 antibody is unknown, we speculated that 55B11 antibody would react with the intracellular region of some tyrosine kinase molecules, based on the facts that the 55B11 but not the V-85 antibody stained eccrine sweat glands in skin tissues and that various tyrosine kinase inhibitors, including Apatinib (a VEGFR2 tyrosine kinase inhibitor), cause limb syndrome due to eccrine sweat gland injury [28]. Therefore, VEGFR2 expression analysis using the V-85 antibody in normal and tumor tissues (those stained with the 55B11 antibody) would be required to properly predict the safety of our TACTICs approach and potentially expand its indication to other solid tumors. To achieve this, we may need to follow an approach that combines in vivo administration of the labeled V-85 antibody and imaging diagnosis. Recently, it has been suggested that the therapeutic effect of VEGF/VEGFR2 inhibition in colorectal cancer patients, using an anti-VEGFR2 antibody (Ramucirumab), is linked to the concentration of VEGF-D in peripheral blood [29]. In the future, tumor environment prediction via liquid biopsies might be possible, reducing the physical burden of invasive biopsy procedures on patients and the risk of cancer cell dissemination. Furthermore, the development of such liquid biopsy-based methods would be advantageous for the determination of both the timing and frequency of mRNA-transfected CAR-T cells administration in our TACTICs therapy. Based on all of the abovementioned, it is essential to estimate the possible sites of adverse events caused by TACTICs therapy and develop countermeasures to safely conduct and execute clinical studies. If we follow this rationale, we believe that a future FIH study using our TACTICs approach will demonstrate both its safety and effectiveness.

## 4. Materials and Methods

### 4.1. Cells

Murine L1.2 cells were kindly provided by Prof. Takashi Nakayama (Kindai University, Higashiosaka, Japan) and cultured in RPMI 1640 medium supplemented with 10% fetal bovine serum (FBS, Thermo Fisher Scientific, Waltham, MA, USA). The VEGFR2^+^ L1.2 cells were generated via retroviral vector transduction (containing the VEGFR2 gene and a puromycin resistance cassette) and were grown in the same culture medium as untransduced L1.2 cells, supplemented with puromycin. NIH/3T3 cells were obtained from the Japanese Collection of Research Bioresources Cell Bank (Osaka, Japan). The VEGFR2^+^ NIH/3T3 cells were also generated via retroviral vector transduction. These NIH/3T3 cell lines were cultured in Dulbecco’s modified Eagle’s medium supplemented with 10% FBS. Human Jurkat cells were obtained from the Cell Resource Center for Biomedical Research, Institute of Development, Aging, and Cancer, Tohoku University (Sendai, Japan). The CAR^+^ Jurkat cells were also generated via transduction, using a retroviral vector containing the CAR#1 and GFP gene. These Jurkat cell lines were cultured in RPMI 1640 medium supplemented with 10% FBS. All cells were maintained in a humidified atmosphere of 5% CO_2_ at 37 °C.

### 4.2. Construction of CAR-mRNA Variants

The CAR-mRNA constructs used in this study are summarized in Figure 1A and Appendix A. The CAR [V/28/28/28-3z] (CAR#1) and CAR [V/8a/8a/137-3z] constructs [16] share the same Igκ-chain leader sequence, HA-tag, anti-VEGFR2 scFv (clone V-85) [30,31], His-tag, and Flag-tag. CAR#1 has CD28-derived HD/TMD and CD28- and CD3ζ-derived STDs. CAR [V/8a/8a/137-3z] has CD8α-derived HD/TMD and CD137- and CD3ζ-derived STDs. Each construct was subcloned into pcDNA3.1-Zeo (Thermo Fisher Scientific, Waltham, MA, USA). CAR#2 (without the various tag sequences) and CAR#3 (codon-optimized) constructs were generated using artificial gene synthesis (Integrated DNA Technologies, Coralville, IA, USA) and were subcloned into pcDNA3.1-Zeo. Sequence integrity of all plasmids was confirmed using DNA sequencing (Fasmac Co., Atsugi, Japan). Using these plasmids as templates, CAR-mRNAs (#1–#3) containing ARCA as the 5′ cap structure were prepared using the mMESSAGE mMACHINE T7 Ultra kit (Thermo Fisher Scientific, Waltham, MA, USA). CAR#4- or CAR#5-coding mRNAs containing CleanCap as the 5′ cap structure were synthesized using wild-type uridines (CAR#4) or 5-methoxyuridines (CAR#5), considering CAR#3 as the gene template, via Trilink (San Diego, CA, USA).

### 4.3. Production of CAR-T Cells

Human CAR-T cells were produced as previously described [16,32]. Briefly, human T cells derived from healthy volunteers were activated using an anti-CD3 mAb (Janssen Pharmaceutica, Beerse, Belgium) for 13 days, and CAR-mRNAs were introduced into 1 × 10^7^ T cells via electroporation using the MaxCyteGT system (MaxCyte, Gaithersburg, MD, USA). Of note, we have confirmed that almost all of the EP-treated T cells were recovered with a high survival rate (cell recovery rate, 90–100%; viability, >97%). The prepared CAR-T cells were maintained in ALyS505N-175 (Cell Science & Technology Institute, Tokyo, Japan) under a humidified atmosphere of 5% CO_2_ at 37 °C.

For the study of CAR-T cell freezing, CAR#1-T cells cultured for 3 h after EP treatment were mixed with CP-1 (Kyokuto Pharmaceutical Industrial, Tokyo, Japan) and slowly frozen using a programmed freezer. The frozen CAR-T cells were then rapidly thawed using a water bath set at 37 °C. For the investigation of CAR-T cell preservation under different temperature conditions, CAR#4-T cells cultured for 18 h were stored in a cold storage bag along with a temperature logger at 4 or 18 °C. After 6 h, the cells were collected and subjected to experiments. The viability of each cell line was assessed using the trypan blue exclusion test.

### 4.4. Reverse Transcription-Quantitative Polymerase Chain Reaction

Total RNA was isolated from CAR-T cells with the TRIzol reagent (Thermo Fisher Scientific, Waltham, MA, USA), and reverse-transcribed using the SuperScript III Reverse Transcriptase (Thermo Fisher Scientific, Waltham, MA, USA). CAR cDNA was detected using Custom TaqMan Gene Expression Assay (Thermo Fisher Scientific, Waltham, MA, USA) targeting the anti-VEGFR2 scFv. CAR and GAPDH expression levels were measured using the CFX96 Real-Time PCR Detection System (Bio-Rad Laboratories, Hercules, CA, USA). CAR-mRNA level was normalized to the GAPDH-mRNA level based on the comparative threshold cycle method (2^−ΔCt^).

### 4.5. Flow Cytometry Analysis: CAR Surface Expression and VEGFR2-Binding

Human CAR-T cells were incubated with Human TruStain FcX (Biolegend, San Diego, CA, USA) and then stained with a LIVE/DEAD Fixable Aqua Dead Cell Stain Kit (Thermo Fisher Scientific, Waltham, MA, USA) and with PE-Cy7-labeled anti-CD8α mAb (clone HIT8α, Biolegend, San Diego, CA, USA). For CAR expression analysis based on binding to VEGFR2 protein, cells were further stained with recombinant human VEGFR2/KDR Fc chimera protein (VEGFR2-Fc, R&D Systems, Minneapolis, MN, USA) and APC-labeled anti-human IgG Fc mAb (clone HP6017, Biolegend, San Diego, CA, USA). For CAR expression analysis using HA-tag detection, cells were further stained with APC-labeled anti-HA-tag mAb (clone 16B12, Biolegend) or APC-labeled mouse IgG1 isotype control mAb (clone MOPC-21, Biolegend, San Diego, CA, USA). Cells were resolved using BD FACS Canto II (BD Biosciences, Franklin Lakes, NJ, USA) or Gallios (Beckman Coulter, Brea, CA, USA) flow cytometers, and data were analyzed using FlowJo software v10 (FlowJo LLC, Ashland, OR, USA). Data were represented as GMFI ratios, calculated according to the following formula: GMFI ratio = (GMFI of CAR-T cells stained with anti-HA mAb or anti-human IgG Fc mAb with VEGFR2-Fc)/(GMFI of CAR-T cells stained with mouse IgG1 isotype control mAb or anti-human IgG Fc mAb without VEGFR2-Fc).

### 4.6. In Vitro Cytotoxicity Assay

L1.2 cells were labeled with Tag-It Violet Proliferation Cell Tracking Dye (Biolegend, San Diego, CA, USA), and VEGFR2^+^ L1.2 cells were labeled with Cell Proliferation Dye eFluor 670 (Thermo Fisher Scientific, Waltham, MA, USA). mRNA-transfected CAR-T cells 24 h after EP, L1.2 cells, and VEGFR2^+^ L1.2 cells were co-cultured at the indicated effector cells-to-VEGFR2^+^ L1.2 cells (E/T) ratios. After 18 h, CountBright Absolute Cell Counting Beads (Thermo Fisher Scientific, Waltham, MA, USA) and 7-AAD Viability Staining Solution (Biolegend, San Diego, CA, USA) were added to the reaction wells, and the number of each target cell was analyzed using flow cytometry until 1000 beads were detected. Cytotoxicity was calculated using the following formula: percentage of VEGFR2-specific lysis = [(VEGFR2^+^ L1.2 cells/L1.2 cells ratio in the non-effector cells’ well) − (VEGFR2^+^ L1.2 cells/L1.2 cells ratio in the effector cells’ well)]/[VEGFR2^+^ L1.2 cells/L1.2 cells ratio in the non-effector cells’ well] × 100.

### 4.7. In Vivo Cytotoxicity Assay

Female NOG mice (NOD/Shi-scid, IL-2RγΚO Jic) were purchased from In-Vivo Science (Tokyo, Japan) and were maintained in the experimental animal facility at Osaka University. The care and use of laboratory animals complied with the guidelines and policies of the Act on Welfare and Management of Animals in Japan. Protocols and procedures were approved by the Animal Care and Use Committee of Osaka University.

A mixture of 2 × 10^6^ VEGFR2^+^ L1.2 cells pre-stained with the Cell Proliferation Dye eFluor 670 and 1 × 10^6^ L1.2 cells pre-stained with the Tag-It Violet Proliferation Cell Tracking Dye were injected into the orbital venous plexus of NOG mice. Two days later, 1 × 10^7^ CAR#4-T cells or Mock-T cells (6 h after 4 °C preservation) were administered to NOG mice via the same route. Two days later, the NOG mice were euthanized and VEGFR2^+^ L1.2 cells and L1.2 cells in their spleens were detected using a flow cytometer. In vivo killing activity of CAR-T cells against VEGFR2^+^ L1.2 cells was calculated using the following formula: percentage of VEGFR2-specific in vivo cytotoxicity = [(VEGFR2^+^ L1.2 cells/L1.2 cells ratio in the vehicle group) − (VEGFR2^+^ L1.2 cells/L1.2 cells ratio in the experimental group)]/[VEGFR2^+^ L1.2 cells/L1.2 cells ratio in the vehicle group] × 100.

### 4.8. Generation of an Antibody Recognizing the Same VEGFR2 Extracellular Domain as CAR

A rabbit antibody (V-85 antibody) with the same variable region as that of anti-VEGFR2 scFv in CAR was obtained from Syd Labs (Boston, MA, USA). To evaluate the VEGFR2 binding properties of this antibody, we performed immunofluorescence and flow cytometry analysis using VEGFR2^+^ NIH/3T3 cells.

For immunofluorescence, NIH/3T3 cells and VEGFR2^+^ NIH/3T3 cells cultured on glass slides were fixed in PBS containing 4% paraformaldehyde and then stained with the V-85 antibody and Alexa Fluor 647-goat anti-rabbit IgG (H + L) highly cross-adsorbed secondary antibody (Thermo Fisher Scientific), or anti-human CD309 (VEGFR2) antibody (clone 7D4-6, Biolegend, San Diego, CA, USA) and Alexa Fluor 647-goat anti-mouse IgG (H + L) highly cross-adsorbed secondary antibody (Thermo Fisher Scientific). Afterward, the antibody solution was thoroughly removed, and the preparations were sealed with ProLong Diamond Antifade Mountant with DAPI (Thermo Fisher Scientific, Waltham, MA, USA). Fluorescence images were acquired using a BZ-X800 fluorescence microscope (Keyence Corporation, Osaka, Japan). For flow cytometric analysis, cells were suspended, stained with the same set of antibodies, and analyzed using a BD FACS Canto II flow cytometer.

### 4.9. Cell-Binding Assay Using VEGFR2^+^ NIH/3T3 and CAR^+^ Jurkat Cells

Jurkat or CAR^+^ Jurkat cells were labeled with Cell Proliferation Dye eFluor 670. 1 × 10^6^ Jurkat cells (per condition) were then added to NIH/3T3 or VEGFR2^+^ NIH/3T3 cells fixed in sheets on glass slides and incubated at 37 °C for 6 h. The Jurkat cell suspensions were thoroughly removed and then the preparations were sealed with ProLong Diamond Antifade Mountant with DAPI. Fluorescence images were acquired using a BZ-X800 fluorescence microscope, and then, the Jurkat cells on the slide were counted for four fields of view. In the competitive inhibition assay using these cell lines (VEGFR2^+^ NIH/3T3 cells and CAR^+^ Jurkat cells), NIH/3T3 cells were first incubated with two anti-VEGFR2 antibodies (V-85 or 7D4-6; 25 ng/mL) overnight at 4 °C, and only then co-cultured with Jurkat cells.

### 4.10. Immunofluorescence Staining of Frozen Tissue Specimens

This study was conducted after receiving approval from the Observational Research Ethics Review Committee of the Osaka University Hospital. STS and surrounding normal tissue (muscle, adipose or skin) blocks were obtained after extensive resections from STS patients and were fixed via immersion in PBS containing 4% paraformaldehyde for 24 h. Preparation of frozen tissue blocks, sectioning and hematoxylin and eosin (H&E) staining were performed by the Applied Medical Research Laboratory (Osaka, Japan).

Frozen tissue specimens were blocked with PBS containing 5% goat serum and 0.3% Triton X-100 at room temperature for 1 h. The blocking buffer was removed, and the anti-VEGFR2 mAb (V-85 antibody or 55B11 antibody (Cell Signaling Technology, Danvers, MA, USA)) as well as the purified anti-human CD31 antibody (clone WM59, BioLegend, San Diego, CA, USA) in antibody dilution buffer (PBS containing 1% BSA and 0.3% Triton X-100) were added and allowed to react overnight at 4 °C. After removal of the antibody solution, preparations were thoroughly washed with TBS containing 0.05% Tween-20. Then, the Alexa Fluor 488-goat anti-mouse IgG (H + L) highly cross-adsorbed secondary antibody (Thermo Fisher Scientific, Waltham, MA, USA) and the Alexa Fluor 647-goat anti-rabbit IgG (H + L) highly cross-adsorbed secondary antibody mixture was prepared in antibody dilution buffer, added to the preparations, and allowed to react for 2 h at room temperature. Again, the antibody solution was thoroughly removed, and the preparations were sealed with ProLong Diamond Antifade Mountant with DAPI. Fluorescence images were acquired using a BZ-X800 fluorescence microscope and the brightness (MFI) of each protein detection was measured using the BZ-H4M/Measurement application. Data were represented as ΔMFI of tissues stained with primary and corresponding secondary antibodies to each antigen, minus MFI of tissues stained with secondary antibodies only.

### 4.11. Immunohistochemical Staining of Paraffin-Embedded Specimens

This study was conducted after receiving approval from the Observational Research Ethics Review Committee of the Osaka University Hospital and Hokkaido University Hospital. Tissue microarray blocks were constructed using a manual tissue microarrayer (JF-4; Sakura Finetek Japan, Tokyo, Japan) with a 2.0-mm diameter needle, from two representative tumor areas (both the invasive front and the bulk of the tumor) and from one representative area of non-neoplastic bile duct as an internal control. The finalized array blocks were sliced into 4-μm-thick sections and mounted onto glass slides. Tissue sections were deparaffinized in xylene and rehydrated through a series of graded ethanol solutions. Heat-induced antigen retrieval was carried out in high-pH antigen retrieval buffer (Dako Cytomation, Glostrup, Denmark). Endogenous peroxidase was quenched with 3% H_2_O_2_ for 5 min. These sections were visualized using the HRP-labeled polymer method (EnVision FLEX system, Dako Cytomation). Immunostained sections were counterstained with hematoxylin, dehydrated in ethanol, and cleared in xylene.

### 4.12. Statistical Analysis

The results of the CAR expression profile and cytotoxicity analysis and cell-binding assay are represented as the mean ± SD. Statistical analysis of these data was performed using Student’s *t*-test for a two-group comparison and Dunnet’s test or Tukey’s test for multi-group comparisons using GraphPad Prism 8 software (GraphPad Software, San Diego, CA, USA). The results of in vivo cytotoxic assay are represented ± SE, and their statistical analysis was performed using Welch’s *t*-test. A *p* value lower that 0.05 was considered statistically significant.

## 5. Conclusions

In the present study, we demonstrated that anti-VEGFR2 CAR-T cells prepared using GMP-compatible EP, and a relevant mRNA sequence containing CleanCap as the 5′ cap structure, were highly cytotoxic. Furthermore, we also determined that 4 °C preservation conditions would allow the anti-VEGFR2 CAR-T cells adequate transport and delivery to patients without loss of function. Moreover, ex vivo staining of human samples suggests that our TACTICs therapy targeting VEGFR2 is expected to exert broad-spectrum cytotoxic activity against STSs. Still, some reactivity was also detected against normal tissues, which must be considered in the post-treatment monitoring phase. Based on our results, we hope to conduct an FIH study using TACTICs therapy in the near future, soon after obtaining further safety data using the V-85 antibody and establishing a GMP/GCP-compatible standard operating procedure for CAR-T cell therapy.

## Figures and Tables

**Figure 1 cancers-12-02735-f001:**
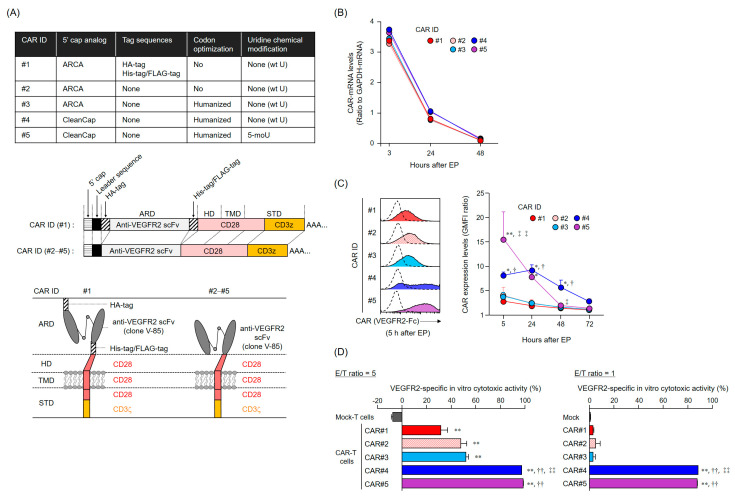
Chimeric antigen receptor (CAR) expression profile and vascular endothelial growth factor receptor 2 (VEGFR2)-specific cytotoxic activity of human T cells transfected with CAR-mRNA variants using electroporation (EP). (**A**) List of CAR constructs used in this study. The upper table shows the details of the five VEGFR2-specific CAR [V/28/28/28-3z]-mRNA constructs, with respect to modified 5′ cap structure, tag sequences, codon optimization, and uridine chemical modification. For these CARs, an illustration of CAR-mRNA is shown in the middle panel and an illustration of CAR proteins is shown in the bottom panel. Abbreviations: ARCA, anti-reverse cap analog; ARD, antigen recognition domain; HD, hinge domain; STD, signal transduction domain; TMD, transmembrane domain; wt U, wild type uridine; 5-moU, 5-methoxyuridine. (**B**) Profile of CAR-mRNAs introduced into human T cells. CAR-mRNAs introduced into human T cells were analyzed using reverse transcription-quantitative polymerase chain reaction. CAR-mRNA expression is represented, relative to that of glyceraldehyde 3-phosphate dehydrogenase (GAPDH)-mRNA. The data are shown as the mean ± standard deviation (SD) of triplicates and are representative of at least two independent experiments using different donor-derived T cells. Statistical analysis was performed using the Dunnett’s test for multiple comparisons with CAR#1 and showed no significant differences. (**C**) Expression profiles of CAR [V/28/28/28-3z] proteins on T cell membranes, translated from the different CAR-mRNA constructs. CAR protein expression was analyzed using flow cytometry using VEGFR2-Fc chimera/anti-human IgG antibody (solid color histograms) or anti-human IgG antibody only (dashed white histograms). CAR expression levels are given using the ratio between VEGFR2-Fc chimera/anti-human IgG staining-geometric mean fluorescence intensity (GMFI) and anti-human IgG antibody staining-GMFI. The histograms on the left are representative results of CAR expression levels, obtained 5 h after EP. The data on the right shows the mean ± SD of three individual experiments using different donor-derived T cells. Statistical analysis was performed using the Tukey’s test: * *p* < 0.05 and ** *p* < 0.01, CAR#4 and CAR#5 versus CAR#1; † *p* < 0.05, CAR#4 versus CAR#3; ‡ *p* < 0.05 and ‡‡ *p* < 0.01, CAR#5 versus CAR#4. (D) VEGFR2-specific in vitro cytotoxic activity of T cells transfected with the different CAR-mRNA constructs. Human CAR-T cells, cultured for 24 h after EP, were co-cultured with L1.2 cells and VEGFR2^+^ L1.2 cells at effector cell/target cell (E/T) ratios of 5 or 1 for 18 h. Then, the number of L1.2 and VEGFR2^+^ L1.2 cells in the wells was evaluated using flow cytometry and the VEGFR2-specific cytotoxic activity was calculated from the ratio of the number of VEGFR2^+^ L1.2 cells to the number of L1.2 cells. The data are represented as the mean ± SD of the three individual experiments using different donor-derived T cells. Statistical analysis was performed using the Tukey’s test: ** *p* < 0.01, CAR-T cells versus mock-T cells; †† *p* < 0.01, CAR#4-T cells and CAR#5-T cells versus CAR#1-T cells; ‡‡ *p* < 0.01, CAR#4-T cells versus CAR#3-T cells.

**Figure 2 cancers-12-02735-f002:**
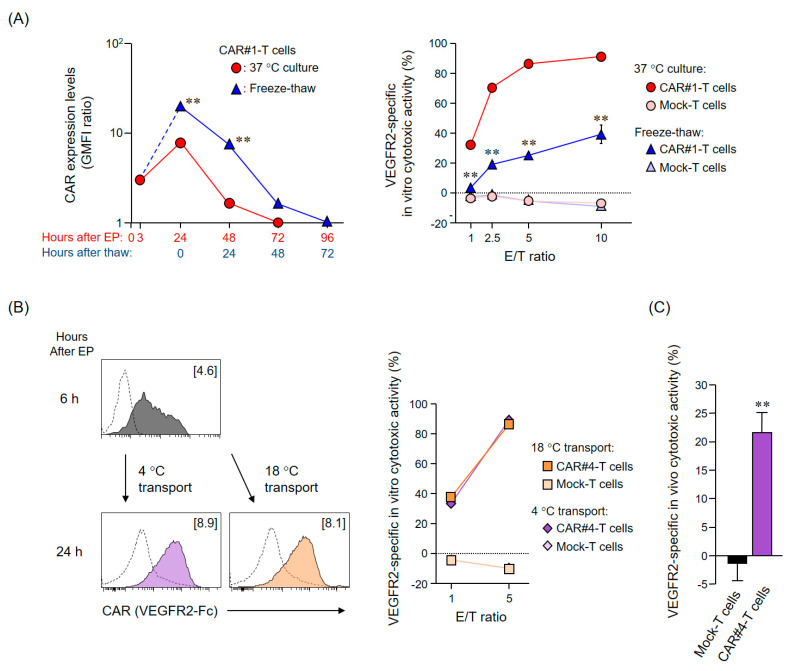
Effects of cryopreservation and cold or room temperature conditions on CAR-mRNA-transfected T cells. (**A**) Effect of cryopreservation on mRNA-transfected CAR-T cells. CAR#1-T cells were cultured for 3 h after EP and then slowly frozen using a frost protection reagent (CP-1). Twenty-one hours later, they were thawed rapidly (24 h after EP). Then, CAR expression (left panel) and VEGFR2-specific in vitro cytotoxic activity (right panel) were evaluated in freeze-thawed CAR-T cells and compared to those of continuously cultured CAR-T cells at 37 °C for 24 h after EP. CAR expression was analyzed using flow cytometry using an anti-HA-tag antibody or the respective isotype control antibody. The VEGFR2-specific in vitro cytotoxic activity of CAR-T cells was evaluated via co-culture with L1.2 cells and VEGFR2^+^ L1.2 cells at the indicated E/T ratios. The data are shown as the mean ± SD of triplicates and are representative of at least two independent experiments using the same donor-derived T cells. Statistical analysis was performed using the Student’s *t*-test: ** *p* < 0.01, freeze-thawed CAR#1-T cells versus 37 °C cultured CAR#1-T cells. (**B**) Effect of different temperature conditions on mRNA-transfected CAR-T cells. Human T cells transfected with CAR#4-coding mRNA via EP were cultured for 18 h and then subjected to cold (4 °C) or room temperature (18 °C) conditions for 6 h. Then, CAR expression (left panel) and VEGFR2-specific in vitro cytotoxic activity (right panel) were evaluated. CAR expression was analyzed using flow cytometry using VEGFR2-Fc chimera/anti-human IgG antibody (solid color histograms) or anti-human IgG antibody only (dashed white histograms). The VEGFR2-specific in vitro cytotoxic activity was evaluated via co-culture with L1.2 cells and VEGFR2^+^ L1.2 cells at the indicated E/T ratios. The data are shown as the mean ± SD of triplicates and are representative of at least two independent experiments using different donor-derived T cells. (**C**) Analysis of in vivo cytotoxic activity of mRNA-transfected CAR-T cells preserved under 4 °C conditions. NOG mice received fluorescently labeled L1.2 and VEGFR2^+^ L1.2 cells via retro-orbital vein injection. Two days later, mice received CAR#4-T cells or mock-T cells preserved at 4 °C via the same route. Then, two days later (four days after L1.2 and VEGFR2^+^ L1.2 cells administration), the number of L1.2 cells and VEGFR2^+^ L1.2 cells in the spleen of these mice was evaluated using flow cytometry. The VEGFR2-specific in vivo cytotoxic activity was given using the ratio of VEGFR2^+^ L1.2 to L1.2 cells. The data are shown as the mean ± standard error of the mean of the results obtained for the 4 and 6 mice treated with mock-T and CAR#4-T cells, respectively. Statistical analysis was performed using the Welch’s t-test: ** *p* < 0.01.

**Figure 3 cancers-12-02735-f003:**
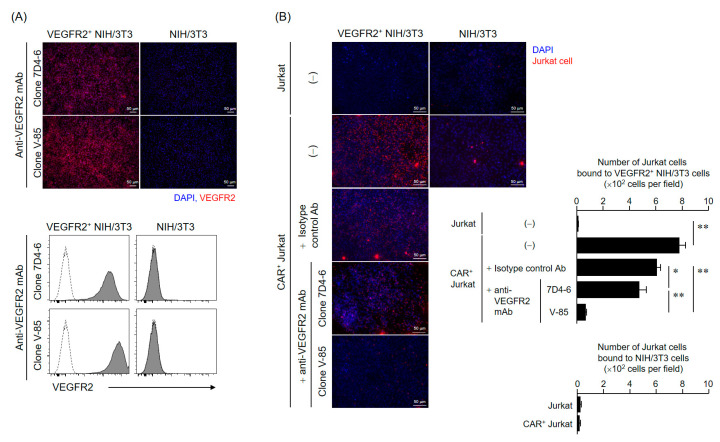
V-85 antibody binding to VEGFR2. A complete antibody possessing the variable region of anti-VEGFR2 scFv (clone V-85), the one used as the antigen recognition domain of CAR, was generated, and its binding potential was evaluated. (**A**) Evaluation of V-85 antibody binding to VEGFR2 using immunofluorescence staining (top panel) and flow cytometry analysis (bottom panel) using VEGFR2^+^ NIH/3T3 and NIH/3T3 cells. Immunofluorescence staining of VEGFR2^+^ NIH/3T3 cells and NIH/3T3 cells was performed using a commercial antibody (7D4-6) recognizing the VEGFR2 extracellular region and Alexa Fluor 647-labeled anti-mouse IgG antibody, or V-85 antibody and Alexa Fluor 647-labeled anti-rabbit IgG antibody. Scale bars = 50 µm. Flow cytometry analyses were performed using the two anti-VEGFR2 antibodies (solid color histograms) and a second antibody only (dashed white histograms). (**B**) Competition assay using VEGFR2^+^ NIH/3T3 cells and anti-VEGFR2 CAR^+^ Jurkat cells to prove the recognition epitope of the V-85 antibody in VEGFR2 is identical to CAR. Fluorescently labeled Jurkat or CAR^+^ Jurkat cells were added to glass plates seeded with VEGFR2^+^ NIH/3T3 or NIH/3T3 cells and cultured for 6 h. The anti-VEGFR2 antibody-added groups were allowed to react with excess amounts of 7D4-6 or V-85 antibody (25 ng/mL) overnight before the addition of the Jurkat cells. The cells that were not strongly bound to NIH/3T3 and VEGFR2^+^ NIH/3T3 cells were washed out. The cells that remained bound to NIH/3T3 and VEGFR2^+^ NIH/3T3 cells were counted using fluorescence microscopy. The data are shown as the mean ± SD of four independent fields. Statistical analysis was performed using the Tukey’s *t*-test: * *p* < 0.05, ** *p* < 0.01.

**Figure 4 cancers-12-02735-f004:**
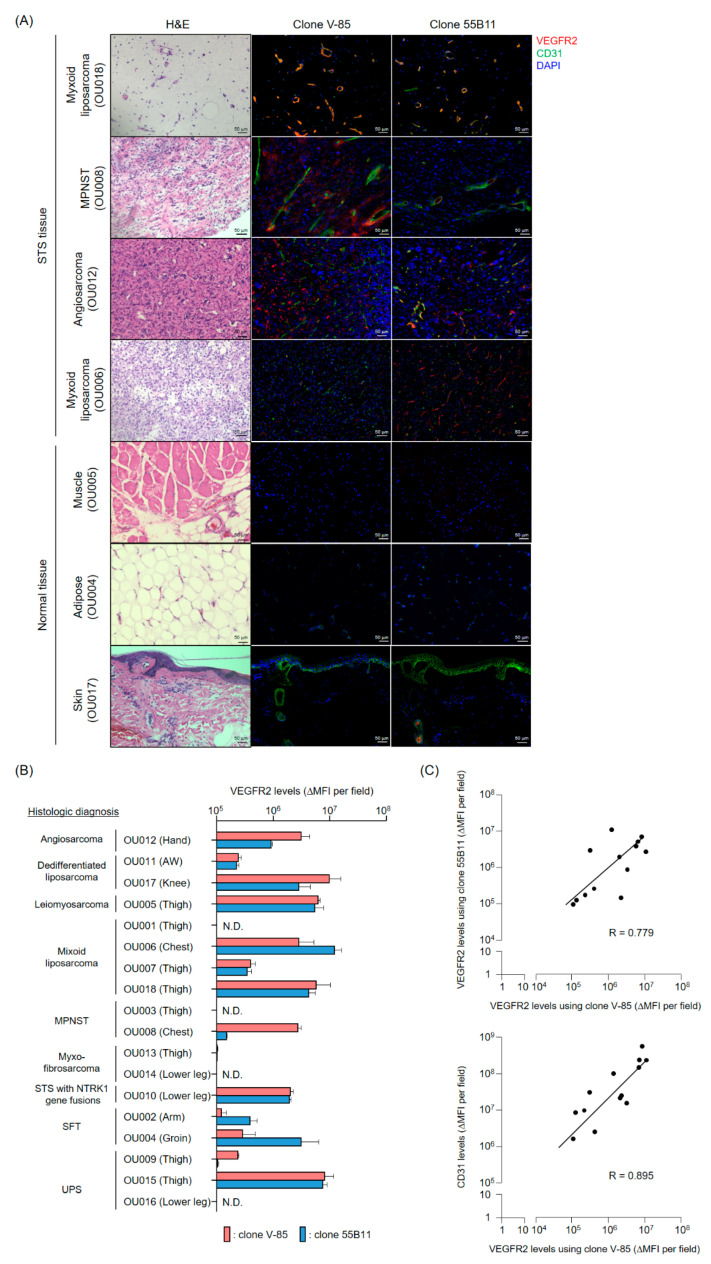
VEGFR2 expression in frozen sarcoma tissues and surrounding normal tissues using immunofluorescence, using the V-85 and 55B11 antibodies. (**A**) Representative images of hematoxylin and eosin (H&E) staining and immunofluorescence staining with anti-VEGFR2 antibody (V-85 or 55B11; red), anti-CD31 antibody (green), and 4′,6-diamidino-2-phenylindole (DAPI) as a counterstain in serial tissue sections of frozen tumor tissues and surrounding normal tissues from various sarcoma patients. Scale bars = 50 µm. (**B**) Expression levels of VEGFR2 in STS frozen tissues. From immunofluorescence staining, the VEGFR2 levels given by the two anti-VEGFR2 antibodies in 18 sarcoma tissues were quantified using a fluorescence microscopy software. The fluorescence intensity of VEGFR2 is shown as the ΔMFI specific for VEGFR2 (corrected for the signal obtained with the secondary antibody alone—Alexa Fluor 647 conjugated anti-rabbit IgG). The data are shown as the mean ± SD of four independent fields. (**C**) Correlation between the V-85-derived VEGFR2 staining (ΔMFI) and the 55B11-derived VEGFR2 staining (ΔMFI, upper panel) or the tumor vessel staining with anti-CD31 antibody (ΔMFI, lower panel). The data are presented for 13 STS cases with VEGFR2 and CD31 expression. Each symbol represents the average of the four fields of view considered in (**B**). Abbreviations: MPNST, malignant peripheral nerve sheath tumor; NTRK1, neurotropic tropomyosin receptor kinase 1; SFT, solitary fibrous tumor; UPS, undifferentiated pleomorphic sarcoma.

**Figure 5 cancers-12-02735-f005:**
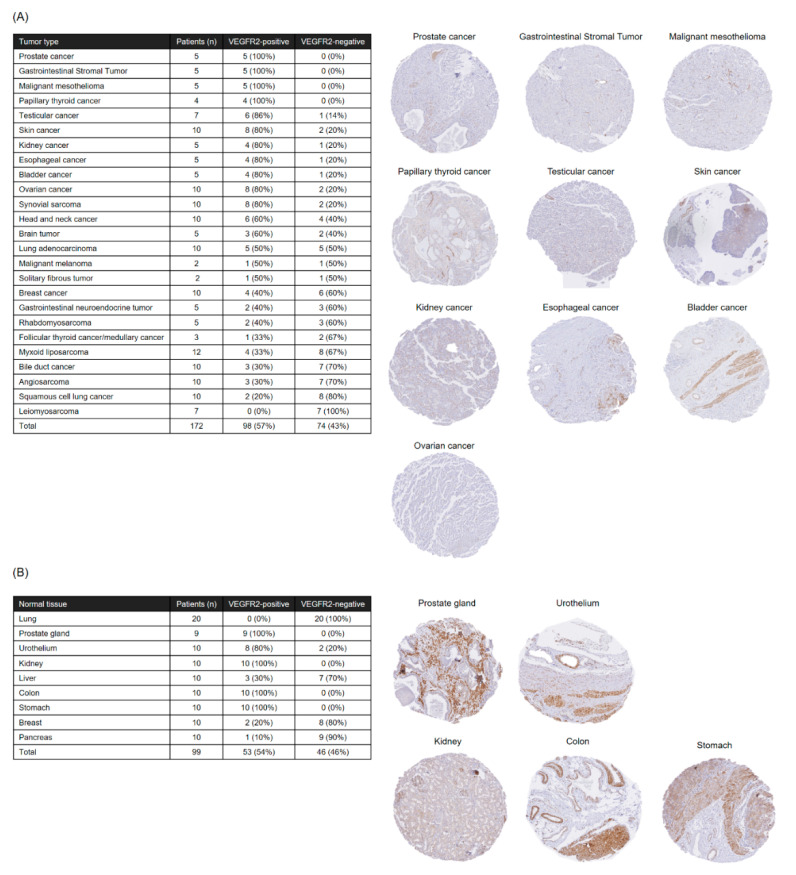
Analysis of VEGFR2 expression in formalin-fixed, paraffin-embedded specimens of various solid tumors (**A**) or normal tissues (**B**) using immunohistochemistry and the 55B11 antibody. The table summarizes the results of immunohistochemistry staining for each cancer type or normal tissue; tissues with a high frequency of VEGFR2 expression are shown as representative images.

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
