# Peer review of "Predicting the Efficacy and Safety of TACTICs (Tumor Angiogenesis-Specific CAR-T Cells Impacting Cancers) Therapy for Soft Tissue Sarcoma Patients"

_cancers, 2020, doi:10.3390/cancers12102735_

Round 1

Reviewer 1 Report

The goal of manuscript by Dr. Fujiwara and colleagues was to optimize the creation of human CAR T cells that target VEGFR2 expressing cells for FIH treatment of patients diagnosed with soft tissue sarcomas (STS).  Because vascular endothelial cells uniformly express VEGFR2, a major concern is destruction of normal vessels by long lived CAR -T targeting VEGFR2. The author previously developed CAR-T cells using viral vectors (refs 15, 16) and more recently changed tack to create CAR- T cells using electroporation of mRNA, which is proposed to limit the in vivo half-life of VEGFR2 CAR expression in CTLs albeit causing cell activation. The rationale is to limit the lifetime of these VEGFR2 targeting CTL cells to limit off-target effects in patients.

The authors’ defined an optimal construct (e.g. no HA seq, Clean cap substituted), demonstrated its killing capacity in vitro and in vivo, and transportation conditions needed to provide optimized cells to patients. In addition, foreseeing a need to detect the presence and level of expression of VEGFR2 in patient derived tissues, the authors developed an antibody to VEGFR2 that was derived from the chimeric VEGFR2 sequence (V-85). This reagent gave similar results to a commercially available anti-VEGFR2 (epitope unknown) used in other cancer research, unfortunately, V-85 Ab did not stain formalin-fixed tissues, the most commonly available, requiring staining frozen tissue.  In summary, this is a well-controlled and well written manuscript that provides important data on VEGFR2 CAR- T cells for potential first in human trials.

 Suggestion for authors:

One comment is that the in vivo cytotoxicity data presented in Fig 2c were very limited in length of time the mice were studied. Hence off target effects may not have appeared (yet). Two point would improve the project: (1)  to determine the half-life of CAR -T VEGFR2+ cells in the in vivo model used; 2) what off target effects, if any, occur during lifetime of VEGFR2 CAR T cells in vivo.

Author Response

Reply to the Review Report (Reviewer 1):

We appreciate you taking the time to offer us your comments related to the paper. We are very grateful that you have a good understanding of our efforts to conduct the FIH study of TACTICs therapy against sarcoma patients that we wanted to convey in this paper. Our response to your comment is as follows

Suggestion;

One comment is that the in vivo cytotoxicity data presented in Fig 2c were very limited in length of time the mice were studied. Hence off target effects may not have appeared (yet). Two point would improve the project: (1) to determine the half-life of CAR -T VEGFR2+ cells in the in vivo model used; 2) what off target effects, if any, occur during lifetime of VEGFR2 CAR T cells in vivo.

Response;

As you point out, we have only been able to conduct limited studies to confirm that the prepared CAR-T cells can injure cancer cells in vivo. It was very difficult to establish an in vivo experimental system in line with the TACTICs therapy strategy because the tumor vessels are constructed with mouse vascular endothelial cells, even in a mouse xenograft model implanted with human tumors. We also attempted to construct tumor-bearing mouse models using either the human VEGFR2-expressing human sarcoma cell line or the human VEGFR2-expressing mouse tumor cell line, but none of these models were able to generate tumors with human VEGFR2 expression on their membrane. For these reasons, we constructed a small-scale, short-term in vivo cytotoxicity assay and obtained only the basic data which would indicate that our final development candidate, anti-VEGFR2 CAR-T cells, could be expected to injure VEGFR2-expressing tumor vessels in clinical trials. Of course, we understand that your two suggestions are important evaluations that are required prior to conducting a clinical trial to determine the number of cells to be administered, the treatment schedule, and to enrich the safety information. In this in vivo study, we found that human T cells on 2 days after T cell administration were present in the spleen, and no CAR protein was expressed on their plasma membranes, suggesting that the CAR-mediated anti-tumor activity of T cells is expected to disappear 1-2 days after administration. In addition, our anti-VEGFR2 scFv (V-85) did not cross-react to mouse VEGFR2, and thus no obvious adverse events were observed in individuals. We plan to evaluate the in vivo functional profiles of clinical-scale CAR-T cells and continue to enhance the safety information of VEGFR2 CAR-T cells by immunohistochemical staining with V-85 antibodies against various normal tissues.

Thank you for your helpful suggestions for future research directions.

Reviewer 2 Report

The authors have previously developed and functionally characterized CAR T cells targeting VEGFR2. As VEGFR2 is known to be overexpressed in tumor vessels, it might be an attractive target for cancer therapy including e.g. sarcomas. In this study, the authors compared different VEGFR2 CAR-coding mRNAs for T cell transfection to find a lead structure for presumptive clinical studies. They further contrasted different preservation conditions of their CAR T cell product and concluded that 4°C transport is the best option. Finally, the suitability of anti-human VEGFR2 Ab V-85 as a diagnostic tool in immunohistochemistry was evaluated.

Major comments

  • In order to reduce immunogenicity of VEGFR2 CAR T cells extracellular tag-sequences were removed. In this regard, it would be important to use also a humanized anti-VEGFR2 scFv. Is the scFv sequence already human/humanized? Is the Ab derived from phage display or is it a rat antibody? Please clarify.
  • As the authors aim to establish a method for producing and transporting larger amounts of CAR T cells for prospective in human studies, it would be relevant to also provide data on T cell numbers and viability before and after EP. How many CAR T cells can be generated in one batch with how much starting material?
  • To increase the clinical relevance, authors should definitely test and compare functionality of their CAR T cells using 2-3 different human VEGFR2+ (sarcoma) cell lines instead of one murine cell line model.
  • To exclude that EP transfection modifies killing properties of T cells, a mock transduced control should be included in the cytotoxicity assays. Please also provide data on (background) killing of human VEGFR2-negative cell lines via VEGFR2 CAR T cells.
  • As CRS is still one major concern in CAR T cell therapy, also cytokine levels should be analyzed and compared in co-cultures of VEGFR2 CAR (#1-5) T cells with VEGFR2-postive or –negative tumors. Please discuss.
  • Does VEGF (ligand) binding impair anti-VEGFR2 CAR T cell functionality?
  • The authors found that CAR #4 is the ”…optimal CAR mRNA construct for the generation of a final development candidate for anti-VEGFR2 CAR-T cell-based therapies”. Therefore, it is not clear why CAR#1 construct was partly used to investigate the optimal transport conditions for mRNA-transfected CAR-T cells. In light of clinical use, this study should be performed only with the final mRNA candidate. Moreover, all conditions (freezing, 4°C, 18°C and 37°C) should be tested in parallel. What might be true for CAR#1 T cells, might be different for CAR#4 T cells.
  • In Figure 1D, CAR1# T cells show the worst cytotoxic activity among all tested constructs with no specific lysis detectable at E:T ratio 1:1. However in Figure 2A, killing was considerably improved. What is the difference between both assays? Does Figure 1D show results of three independent experiment with one or three different T cell donors? Please also show killing data in Figure 2A for three different T cell donors.
  • The authors conclude that transport of mRNA-transfected CAR T cells at 4°C or 18°C does not impact their cytotoxic activity both in vitro and in vivo. However, at an E:T ratio of 1:1 a clear difference in cytotoxicity between 37°C (Figure 1D) and 4°C/18°C conditions (Figure 2B) (~80% versus ~40%) could be observed. As the assays were not conducted in parallel, a direct side-by-side comparison with at least three individual T cell donors is recommended. As only the functionality of “4°C CAR T cells” were assessed in vivo, the author cannot conclude that cytotoxic activity is not affected. Please also include at least CAR T cells cultured at 18°C and 37°C.  
  • Mouse experiment: Why was tumor burden solely measured in spleen? How is the tumor burden in other tissues e.g. bone marrow, blood,..? Why were VEGFR2-pos tumor cells applied alongside with VEGFR2-neg cells in the same mouse? Will a repeated application of CAR T cells improve tumor cell killing over time?
  • It is not clear how the anti-VEGFR2 Ab was generated. What is the origin? Please explain in detail. Was the Ab generated by a company (Syd Labs, as mentoined in the materials/methods) or generated in the authors lab (as mentioned in the results section)?
  • It should be tested whether V-85 has any cross-reactivity towards other VEGFR variants.
  • For assessing the safety profile of VEGFR2 CAR T cells, VEGFR2 expression should be also analyzed (with V-85 Ab) in brain, lung and liver as these organs could be associated with some adverse reactions (that might also be fatal).

Minor comments

  • Page 5, line 182: Please correct “…in cultured;…”
  • Were experiments shown in Figure 1B and 1C conducted with three different T cell donors or are these summarized data of three independent experiments with one T cell donor? It would be important to show that high transfection rates are not only reproducible for one but also for different donors.

Author Response

Reply to the Review Report (Reviewer 2):

We appreciate you taking the time to offer us your comments and insights related to the paper. We found your feedback very constructive and tried to be responsive to your concerns. Our responses are given in a point-by-point manner below. Changes in the manuscript are highlighted in yellow.

Q1;

In order to reduce immunogenicity of VEGFR2 CAR T cells extracellular tag-sequences were removed. In this regard, it would be important to use also a humanized anti-VEGFR2 scFv. Is the scFv sequence already human/humanized? Is the Ab derived from phage display or is it a rat antibody? Please clarify.

A1;

The anti-VEGFR2 scFv, which is the antigen-recognition domain of our CAR-T cells, was obtained from the phage-display scFv library with human VEGFR2-immunized mouse (details are described in Reference 16,28,29) and has not been humanized. We agree that the scFv incorporated into CAR should be humanized to completely eliminate the immunogenicity of CAR-T cells. However, modification of the amino acid sequence of the antibody variable domain may also affect the antigen specificity of scFv and the efficiency of CAR protein expression in T cells. Previous studies have also suggested that the cause of in vivo elimination of administered T cells is not due to heterologous proteins of rodent origin, but rather to chimeric proteins that are not found in nature due to the artificial combination of multiple immune molecules. Therefore, to assess the utility of TACTICs therapy in humans at an early stage, we incorporated anti-human VEGFR2 scFv, obtained by the phage-display method, into CAR without their humanization, while only removing the highly immunogenic tag sequence.

Q2;

As the authors aim to establish a method for producing and transporting larger amounts of CAR T cells for prospective in human studies, it would be relevant to also provide data on T cell numbers and viability before and after EP. How many CAR T cells can be generated in one batch with how much starting material?

A2;

We have previously reported that electroporation using CAR-mRNA is an efficient way to generate CAR-T cells without damaging human T cells. In this study, we have confirmed that CAR-mRNA was introduced into 1x107 T cells by an electroporator, which will be used in clinical trials, and most T cells were recovered with high viability (cell recovery rate, 90-100%; viability, >97%). This information has been added to the Materials and Methods section as follows.

(Page 14, Lines 470-475)

Briefly, ..., and CAR-mRNAs were introduced into the 1 x 107 T cells via electroporation using the MaxCyteGT system (MaxCyte, Gaithersburg, MD, USA). Of note, we have confirmed that most of the EP-treated T cells were recovered with a high survival rate (cell recovery rate, 90-100%; viability, >97%).

Q3;

To increase the clinical relevance, authors should definitely test and compare functionality of their CAR T cells using 2-3 different human VEGFR2+ (sarcoma) cell lines instead of one murine cell line model.

A3;

Our TACTICs therapy has a strategy that targets tumor regression by injuring tumor blood vessels. However, the creation of in vivo animal models for evaluating the efficacy of TACTICs therapy with high clinical relevance is limited because the tumor blood vessels generated in the tumor-bearing mice are composed of mouse vascular endothelial cells, regardless of whether mouse or human cancer cell lines are used. Furthermore, as VEGFR2 has been reported to be expressed in some cancer cell lines, such as synovial sarcoma and melanoma, we confirmed the expression of VEGFR2 in vitro (2D and 3D culture) and in vivo in these cancer cell lines, but unfortunately, no membrane expression of VEGFR2 was observed in any of the cells/tumors. We also generated a mouse cancer cell line expressing human VEGFR2, but these cells did not engraft into wild-type and immunodeficient mice. As we have previously demonstrated the efficacy of TACTICs therapy in syngeneic mouse models, we decided to confirm the cytotoxic activity of the final product candidate, CAR-T cells, against L1.2 cells in vivo blood cancer model.

Q4;

To exclude that EP transfection modifies killing properties of T cells, a mock transduced control should be included in the cytotoxicity assays. Please also provide data on (background) killing of human VEGFR2-negative cell lines via VEGFR2 CAR T cells.

A4;

As you pointed out, we have added the data from Mock-T cells (EP-treated without mRNA) to Figure 1D, 2A, 2B, and Supplementary figure 1D to show that the cytotoxic activity of each CAR-T cell was induced via CAR. In this study, both L1.2 and VEGFR+ L1.2 cells were co-cultured with T cells in a well and the number of cells in each of the two types of L1.2 cells was measured. As shown in Supplementary figure 1D, V/28/28/28-3z construct had no cytotoxic activity against L1.2 cells. Hence, to correct for inter-well error in the comparative analysis of CAR#1-5, we calculated the cytotoxic activity of each T cells from the ratio of VEGFR2 L1.2 cells to L1.2 cells.

Q5;

As CRS is still one major concern in CAR T cell therapy, also cytokine levels should be analyzed and compared in co-cultures of VEGFR2 CAR (#1-5) T cells with VEGFR2-postive or –negative tumors. Please discuss.

A5;

We are well understood that CRS and on-target off-tumor toxicity are the causes of serious adverse events caused by CAR-T cell therapy. On the other hand, it is difficult for us to estimate the risk of CRS in FIH studies from assessing CAR-T cell cytokine secretion in vitro because there is no clear criterion for the serum cytokine levels that trigger CRS occurrence. Therefore, we have chosen the mRNA-EP method as the method of CAR-T cell generation and have a therapeutic regime that prevents more severe side effects compared to CAR-T cells generated by the viral vector gene transfer method, as previously described in reference 16. Although the present study did not evaluate the effect of different CAR constructs on the cytokine secretion of CAR-T cells, the CAR-T cell cytokine secretion is expected to correlate with their cytotoxic activity. In future GLP-compatible non-clinical studies, we plan to analyze the quality and performance of CAR#4-T cells in more detail.

Q6;

Does VEGF (ligand) binding impair anti-VEGFR2 CAR T cell functionality?

A6;

We have not identified the V-85 epitope on the VEGFR2 extracellular domain, and therefore, don’t know whether VEGF-binding to VEGFR2 impairs our CAR-T cell functionality. As you are concerned, we should understand the antigen-binding properties of anti-VEGFR2 CAR-T cells, because the VEGFR2-recognition efficiency of our CAR-T cells may be reduced depending on the position of the V-85 epitope. In future GLP-compatible non-clinical studies, we plan to confirm CAR-T cell functions in the presence of VEGFs as one of the quality and performance tests of CAR-T cells.

Q7;

The authors found that CAR #4 is the ”…optimal CAR mRNA construct for the generation of a final development candidate for anti-VEGFR2 CAR-T cell-based therapies”. Therefore, it is not clear why CAR#1 construct was partly used to investigate the optimal transport conditions for mRNA-transfected CAR-T cells. In light of clinical use, this study should be performed only with the final mRNA candidate. Moreover, all conditions (freezing, 4°C, 18°C and 37°C) should be tested in parallel. What might be true for CAR#1 T cells, might be different for CAR#4 T cells.

A7;

As you point out, conducting an experiment with all cell transport conditions, including cryopreservation, would help to clearly demonstrate the effect of storage conditions on CAR#4-T cells. However, from a scientific perspective, it is unlikely that a slight modification of the CAR construct would have a drastic effect on the freeze-thaw resistance of T cells.

 Since the beginning of the study, we have been working step by step on two approaches to delivering CAR-T cells to patients with a high therapeutic promise: modification of mRNA and investigation of transport conditions. Although cryopreservation of T cells was very promising as a method of their transport and quality control, it was clear from a series of experiments that it greatly diminished the functionality of T cells. In addition, further experiments for data collection would be contrary to human research ethics. Hence, we finally decided to test the effect of transport conditions at 4°C/18°C on the functionality of CAR-T cells.

Q8;

In Figure 1D, CAR1# T cells show the worst cytotoxic activity among all tested constructs with no specific lysis detectable at E:T ratio 1:1. However in Figure 2A, killing was considerably improved. What is the difference between both assays? Does Figure 1D show results of three independent experiment with one or three different T cell donors? Please also show killing data in Figure 2A for three different T cell donors.

A8;

The conditions in those experiments were the same, but the T cell donors used were different. Figure 1D show results in the mean of three independent experiments with different T cell donors, respectively. On the other hand, Figure 2A data are representative of two independent experiments with the same T cell donor. Although the T cells used in the cryopreservation study may have had higher activity than the T cells used in other studies, the impact of cell cryopreservation on CAR-T cell function was so great that we moved on to investigating other transport conditions that could be an alternative to cryopreservation.

Q9;

The authors conclude that transport of mRNA-transfected CAR T cells at 4°C or 18°C does not impact their cytotoxic activity both in vitro and in vivo. However, at an E:T ratio of 1:1 a clear difference in cytotoxicity between 37°C (Figure 1D) and 4°C/18°C conditions (Figure 2B) (~80% versus ~40%) could be observed. As the assays were not conducted in parallel, a direct side-by-side comparison with at least three individual T cell donors is recommended. As only the functionality of “4°C CAR T cells” were assessed in vivo, the author cannot conclude that cytotoxic activity is not affected. Please also include at least CAR T cells cultured at 18°C and 37°C.

A9;

We concluded that transporting mRNA-transfected CAR T cells at 4°C or 18°C did not affect in vitro cytotoxic activity but we did not conclude that their transporting conditions did not affect in vivo cytotoxic activity.

 As mentioned above, we have experienced different results of cytotoxicity assay (especially E/T ratio = 1) depending on the characteristics and the CD8+/CD4+ cell ratio of the donor T cells. In addition, the functionality of CAR-T cells in this experiment may have been affected by temperature and vibration during transport. We speculate that the low cytotoxic activity at E/T ratio = 1 reflects these factors. On the contrary, CAR-T cells transported at 4°C/18°C had the activity to kill most VEGFR2-expressing cells at E/T ratio =5, suggesting that these CAR-T cells continued to have a high therapeutic potential in clinical trials as well as CAR-T cells cultured at 37°C. Here, we evaluated the in vivo function of CAR-T cells transported under 4°C conditions, believing that lower temperature conditions would reduce T cell activity and allow them to be delivered to patients while maintaining high functionality. Therefore, we consider further experiments, including other temperature conditions, to be unnecessary and inappropriate for human research ethics guidelines and in the spirit of animal welfare.

 The interpretation of these data was modified as follows.

(Page 5, lines 201-203)

CAR-T cells subjected to both temperatures showed equivalent high CAR expression levels and cytotoxic activity, comparable to those of CAR-T cells cultured at 37°C for 24 h (Figure 1D and 2B).

Q10;

Mouse experiment: Why was tumor burden solely measured in spleen? How is the tumor burden in other tissues e.g. bone marrow, blood? Why were VEGFR2-pos tumor cells applied alongside with VEGFR2-neg cells in the same mouse? Will a repeated application of CAR T cells improve tumor cell killing over time?

A10;

The purpose of the in vivo cytotoxicity assay in this study was to confirm that mRNA-transfected human CAR-T cells transported at 4°C could injure cancer cells in vivo. As both L1.2 cells and CAR-T cells administered in mice accumulated in the spleen, we analyzed L1.2 cells and VEGFR2-positive L1.2 cells in the spleen and calculated the in vivo cytotoxicity to VEGFR2-positive L1.2 cells. Coadministration of VEGFR2-negative L1.2 cells with VEGFR2-positive L1.2 cells was used to correct for inter-individual analysis errors as well as in vitro cytotoxicity assay.

 Frequent administration of CAR-T cells is expected to eliminate transplanted VEGFR2- positive L1.2 from the body, depending on the number of times the cells are administered, whereas high doses of human cells (xenocytes) to mice induce the development of graft-versus-host disease. We believe that the current data are sufficient for our purposes and that the design of such an experimental system is inappropriate for human research ethics guidelines and in the spirit of animal welfare.

Q11;

It is not clear how the anti-VEGFR2 Ab was generated. What is the origin? Please explain in detail. Was the Ab generated by a company (Syd Labs, as mentoined in the materials/methods) or generated in the authors lab (as mentioned in the results section)?

A11;

For the V-85 antibody, we commissioned Syd labs to synthesize rabbit IgG with the anti-VEGFR2 scFv (V-85) variable region. The relevant sections in the text have been slightly modified as follows.

(Page 7, lines 244-246)

Hence, we obtained a complete antibody possessing the variable region of anti-VEGFR2 scFv (clone V-85), the one used as the antigen recognition domain of CAR, and evaluated its binding properties to VEGFR2.

Q12;

It should be tested whether V-85 has any cross-reactivity towards other VEGFR variants.

A12;

We have obtained the scFv (V-85) with strong binding properties against human VEGFR2 under strict panning conditions, but we have not confirmed any cross-reactivity to VEGFRs other than VEGFR2. Although the results of immunohistochemistry in this study showed no strong binding of the V-85 antibody to muscle, adipose, and skin tissue that express VEGFR1/3, we would like to evaluate the binding properties of V-85 in GLP-compatible non-clinical studies to enrich the safety information of TACTICs therapy.

Q13;

For assessing the safety profile of VEGFR2 CAR T cells, VEGFR2 expression should be also analyzed (with V-85 Ab) in brain, lung and liver as these organs could be associated with some adverse reactions (that might also be fatal).

A13;

We agree with the assessment of VEGFR2 expression and V-85 binding in major tissues such as brain, lung, and liver. On the other hand, the staining of V-85 antibodies can only be examined in frozen tissues, and staining using frozen tissues is greatly affected by the process of freezing and storage of the tissues, and therefore, we have not been able to obtain tissue specimens suitable for examination. Because of these limitations, we will prepare for the conduct of clinical study of TACTICs therapy, while examining the antigen recognition properties of the V-85 antibody against new tissue samples.

Minor comments:

Q14;

Page 5, line 182: Please correct “…in cultured;…”

A14;

Thank you for pointing out the error. We have corrected the line 183 on page 5.

“Interestingly, the CAR expression levels in frozen CAR-T cells, immediately after thawing, was higher than that in the CAR-T cells cultured at 37°C for 24 h; ...”

Q15;

Were experiments shown in Figure 1B and 1C conducted with three different T cell donors or are these summarized data of three independent experiments with one T cell donor? It would be important to show that high transfection rates are not only reproducible for one but also for different donors.

A15;

Figure 1 shows three independent experiments with different donor-derived T cells. Figure 1B is the representative data analyzing CAR-mRNA level introduced into T cells for the first two of these experiments. Including our previous reports (Reference 16), electroporation is a highly efficient way to introduce CAR-mRNA into T cells. Donor information for the T cells used in the experiments in Figure 1/2 was added to each figure legend (Page 4, 6).

Finally, thank you for giving us the opportunity to strengthen our manuscript with your valuable comments and queries. We have worked hard to incorporate your feedback and hope that these revisions persuade you to accept our submission.

Round 2

Reviewer 1 Report

Revisions addressed my concerns.

Author Response

Thank you for your time.

Reviewer 2 Report

Based on my previous comments the authors revised their manuscript. However, the following points should be addressed before publication:

1) Please point out that the scFv of the CAR is not yet humanized (in this manuscript) and discuss/mention the remaining immunogenic risk accordingly. Cite appropriate literature.

2) The authors should verify their in vitro data with a second human cell line. Surface expression of VEGFR2 was shown e.g. on HUVEC cells.

3) All in vitro experiments should be performed and shown for 3 different T cell donors. 

Author Response

We thank you for reviewing our replies and for your further suggestions. Our responses to your latest comments are given in a point-by-point manner below. Changes in the manuscript are highlighted in yellow.

Q1;

Please point out that the scFv of the CAR is not yet humanized (in this manuscript) and discuss/mention the remaining immunogenic risk accordingly. Cite appropriate literature.

A1;

We have added the following text to the discussion in accordance with your suggestion. As we have previously responded, we do not consider that scFv sequence derived from a mouse antibody poses a highly immunogenic risk. If you have evidence that CAR-T cells with mouse scFv have serious immunogenicity, please let us know.

(Page 12, Lines 380-394)

In addition, a series of experiments on CAR-mRNA construct modification revealed that the removal of the tag sequences in the CAR extracellular domain did not affect the efficiency of CAR expression and activation. Reducing the antigenicity of the CAR extracellular domain to the host immune system is an important consideration, as the appearance of human anti-mouse antibody (HAMA) and human anti-CAR antibody (HACA) can diminish the efficacy of CAR-T cell therapy [25]. On the contrary, our anti-VEGFR2 scFv (V-85), which was incorporated as the CAR extracellular domain, is a mouse-derived sequence that has not been humanized. Although we believe that scFv should be humanized to completely eliminate the antigenicity of CAR-T cells, the humanization of scFv carries the risk of altering the efficiency of CAR surface expression and antigen recognition properties [26]. Additionally, clinical trials using non-humanized anti-Mesothelin CAR-mRNA transfected CAR-T cells suggested that the appearance of HACA, but not HAMA, limits the efficacy of CAR-T cell therapy. Hence, in the present study, we sought to improve the CAR-mRNA constructs to limit the risk of HACA appearance [21]. Needless to say, we plan to measure HACA and HAMA against CAR#4-T cells as biomarkers to monitor the efficacy and safety of the TACTICs therapy in the context of the FIH study.

Q2;

The authors should verify their in vitro data with a second human cell line. Surface expression of VEGFR2 was shown e.g. on HUVEC cells.

A2;

We well understand that it is important to confirm that each of the small molecules and biologics shows similar efficacy against several cancer cell lines because these agents aim to induce cancer cell apoptosis by regulating their intracellular molecules and signaling systems and because the tumor is composed of heterogeneous cancer cells. On the contrary, CAR-T cells directly kill cancer cells, and therefore, we thought that it is more important to evaluate the equivalence of CAR-T cells generated from several donor-derived T cells and the number of cancer cells and vascular endothelial cells is not a critical issue for this study. In the present study, we evaluated CAR-T cells generated from at least two different donors. Although the functionalities of CAR-T cells differed slightly between the different donors, all the CAR#4-T cells that we selected as the final product candidate had the ability to exert potent cytotoxic activity at an E/T ratio of 5. Therefore, we believe that our selection of CAR-mRNA constructs and the T-cell transport condition allow for a consistent supply of CAR-mRNA transfected T cells that show high killing activity against tumor endothelial cells.

 We also performed cytotoxicity assays with HUVECs early in this project. However, the cytotoxic activity of CAR-T cells against HUVECs with low levels of VEGFR2 expression was weak, and CAR-independent killing due to differences in MHC haplotype between effector and target cells was strongly observed. Therefore, we decided that targeting human cells, including HUVECs, was not appropriate for assessing the functional characteristics and VEGFR2 specificity (safety) of CAR-T cells and established an experimental system using VEGFR2 high-expression cell lines (NIH/3T3 and L1.2 cells). As the reactivity of CAR-T cells to various antigen densities is an important concern, we have reported in a previous paper that our CAR-T cells exhibit cytotoxic activity in response to VEGFR2 density. In the present study, we show that VEGFR2 expression levels differ between vascular endothelial cells in tumor tissues and those in normal tissues. These data support the possibility that our CAR-T cells do not injure normal vascular endothelial cells, including HUVECs.

 Again, we believe that it is important to evaluate the functionality of T cells generated from different donors in order to advance the development of CAR-T cells. Then, in this study, it was necessary to use the VEGFR2 high expression cell line to evaluate the functionality of our anti-VEGFR2 CAR-T cells.

Q3;

All in vitro experiments should be performed and shown for 3 different T cell donors.

A3;

We also assume that the more experiments with different donor-derived T cells, the more likely it is that the differences of the donor T cell characteristics can be properly determined as individual differences. At the beginning of our research plan, we planned to evaluate the cryopreservation conditions for T cells from three or more donors. However, as described in this paper, the effect of the freeze-thawing procedure on T-cell function was so great that we changed our plan to examine the effects of the low-temperature transport conditions on the T cell. In addition, due to a COVID-19 pandemic, we were unable to collect a blood sample from healthy volunteers as we had planned. Therefore, only two donor-derived T cells were available for in vitro experiments of transport conditions. However, as mentioned above, the series of experiments suggest that the differences in the functionality of the different donor CAR-T cells are insignificant.

 New experiments using T cells from different donors must again be approved by an ethics committee to be conducted. However, in terms of research ethics, they will not easily approve the research plan for data collection experiments, and even if they do, it is likely to take a lot of time to obtain its approval. Therefore, we will further validate the properties and quality of the cells and their transport conditions in GLP-compatible non-clinical studies that are currently being planned.
